# Species-Level Characterization of the Microbiome in Breast Tissues with Different Malignancy and Hormone-Receptor Statuses Using Nanopore Sequencing

**DOI:** 10.3390/jpm13020174

**Published:** 2023-01-19

**Authors:** Lan Luo, Aisi Fu, Manman Shi, Jiawei Hu, Deguang Kong, Tiangang Liu, Jingping Yuan, Shengrong Sun, Chuang Chen

**Affiliations:** 1Department of Thyroid and Breast Surgery, Renmin Hospital of Wuhan University, Wuhan 430060, China; 2Key Laboratory of Combinatorial Biosynthesis and Drug Discovery, Ministry of Education and Wuhan University School of Pharmaceutical Sciences, Wuhan 430060, China; 3Department of Pathology, Renmin Hospital of Wuhan University, Wuhan 430060, China

**Keywords:** microbiome, breast cancer, 16S rRNA, nanopore sequencing, diversity

## Abstract

Unambiguous evidence indicates that microbes are closely linked to various human diseases, including cancer. Most prior work investigating the microbiome of breast tissue describes an association between compositional differences of microbial species in benign and malignant tissues, but few studies have examined the relative abundance of microbial communities within human breast tissue at the species level. In this work, a total of 44 breast tissue samples including benign and malignant tissues with adjacent normal breast tissue pairs were collected, and Oxford Nanopore long-read sequencing was employed to assess breast tissue microbial signatures. Nearly 900 bacterial species were detected from the four dominant phyla: Proteobacteria, Firmicutes, Actinobacteria and Bacteroidetes. The bacteria with the highest abundance in all breast tissues was *Ralstonia pickettii*, and its relative abundance increased with decreasing malignancy. We further examined the breast-tissue microbiome composition with different hormone-receptor statuses, and the relative abundance of the genus *Pseudomonas* increased most significantly in breast tissues. Our study provides a rationale for exploring microbiomes associated with breast carcinogenesis and cancer development. Further large-cohort investigation of the breast microbiome is necessary to characterize a microbial risk signature and develop potential microbial-based prevention therapies.

## 1. Introduction

Breast cancer has become the most prevalent cancer worldwide and the first leading cause of cancer death in women [1]. The occurrence of breast cancer can be attributed to a variety of related risk factors, such as age, family history, alcohol consumption, and obesity [2], but the exact etiology of breast cancer remains undetermined, and the probability of cancer development, treatment effect, and survival still cannot be accurately predicted [3]. There are additional factors associated with the development of breast cancer waiting to be explored, one of which is the human microbiome. Previous studies have revealed that microbial pathogens are etiological factors in approximately 15–20% of cancers worldwide [4], including gastric [5] and colorectal cancer [6]. Furthermore, multiple tumor types were found to have distinct microbial compositions, although the contribution of these microbes to tumorigenesis and cancer progression is only beginning to be explored and merits further mechanistic investigation [7]. It is currently believed that the ways in which microbes contribute to carcinogenesis fall into several categories: directly damaging host cell DNA, altering cell signaling pathways, modulating immune cells and inflammation, and influencing the metabolism of host-produced factors and pharmaceuticals [8].

In recent years, a greater appreciation of the microbes inhabiting female mammary gland sites has emerged. Due to its high percentage of adipose composition with lymphatic drainage and extensive vasculature, the breast may provide a favorable environment for colonization and growth of bacteria [9]. Accumulating studies have found that the human breast harbors a unique and diverse microbiota [10,11], which not only differs between pathologically-defined normal and malignant breast tissues, but also with clear distinctions among race, tumor subtype, and stage of breast cancer [12,13,14,15]. Recent studies have demonstrated correlations between specific microbes and the clinical outcomes in breast cancer [16], and suggested that tumor-resident microbiota play an important role in promoting breast cancer metastasis [17]. Although progress has been made to support the existence and potential function of the breast tissue microbiome, further in-depth exploration is required to identify the diversity and complexity of breast microbiota composition and determine whether the shifts in microbial assemblages link to breast carcinogenesis [18,19]. In addition, certain microbial communities derived from the mouth and gut have been detected in breast tissue, such as *Fusobacterium nucleatum* [20] and *Bacteroides fragilis* [21], which have been shown to promote breast cancer tumor growth and metastasis. Clearly, accurate detection and identification of specific microbial species in the breast are of great importance.

Second-generation sequencing technologies, such as Illumina [22,23] and Ion Torrent [24], have been widely applied in the collection of 16S rRNA sequence data. These platforms provide a wealth of short-read lengths (200–400 bp) with high accuracy (~99%), but also lead to limited species resolution, rarely below genus level [25]. Oxford Nanopore Technologies (Oxford, UK; ONT) has developed a third-generation platform for long-read and direct sequencing of individual strands of DNA [26,27], and the ability to provide a full-length sequence of 16S rRNA genes leads to more accurate identification of microorganisms at the species level [28,29]. Within the last five years, the sequence accuracy of ONT nanopore sequencing has been improved from 60% to >90%, and researchers have begun to use it to characterize the composition of the microbial community in clinical samples, such as human respiratory samples [30], fecal samples [31], and colorectal cancer tumor tissues [32].

In this study, we attempted to investigate the microbiome in human benign and malignant breast tissue (including adjacent normal tissue) using the nanopore sequencing platform, and to determine the bacterial community structures in breast tissue with different hormone-receptor expressions. We sought to address the hypothesis that microbial populations living within the breast display a different relative abundance depending on the benign or malignant state, and that hormone-receptor expression is strongly associated with microbiome profiles. As far as we know, our study is the first to reveal the composition of breast tissue microbes at the species level using near full-length 16S rRNA operon reads. A deeper understanding of the breast microbiome as impacted by disease state and tumor subtype may contribute to facilitating earlier clinical intervention leading to improved breast cancer patient outcomes.

## 2. Materials and Methods

### 2.1. Patients Enrollment and Breast Tissue Collection

The study was approved by the institutional ethics committee of Renmin Hospital of Wuhan University (WDRY2020-K194). The research was conducted in accordance with the Declaration of Helsinki, and followed all relevant guidelines and regulations. Fresh-frozen breast tissue was aseptically collected from 26 women undergoing non-mastectomy breast surgery for cancer or benign disease at Renmin Hospital of Wuhan University in Wuhan, Hubei Province, China. Patients included were identified with no history of antibiotic use within 6 months and no other disease or condition that might have affected the study assessments. Written informed consent was obtained from each participant prior to obtaining breast tissue. Breast cancer (Tumor) tissue and adjacent normal breast tissue pairs (Normal Pair) came from the same donor. “Adjacent normal tissue” is defined as non-malignant breast tissue located up to 5 cm away from the edge of the tumor, and was evaluated and confirmed by a pathologist to be histologically free of any tumor cells or lesions. After resection, samples were immediately placed in sterile, study code-labeled tubes and were frozen in liquid nitrogen, then stored in a −80 °C freezer until use.

### 2.2. Sample Processing and DNA Extraction

All tissue samples were sent to the clinical laboratory for DNA extraction. Tissue samples were cut into small pieces using sterile scissors and homogenized in batches according to their unique group and number. DNA was extracted from 200 μL of pre-treated samples using the Sansure DNA Extraction Kit (Changsha, China) following the manufacturer’s instructions. DNA was also extracted from two samples of 200 μL Tris- EDTA buffer, with each batch of clinical samples as a no-template control.

### 2.3. Amplification and Nanopore Targeted Sequencing

The universal primers 27F/1492R (for bacterial 16S rRNA gene) [33] were optimized and used in nanopore sequencing; the details of primer design and PCR procedure are described elsewhere [34]. Then, the amplification products of 16S rRNA from the clinical sample and negative controls were pooled for sequencing library construction, using the 1D Ligation Kit (SQK-LSK109, Oxford Nanopore Technologies). Sequencing was performed using an R9.4 flow cell on the GridION x5 (Oxford Nanopore Technologies). Sequencing data was generated continuously after sequencing began. In the default script of the sequencer, a fastq file is generated for every 4000 reads, and real-time data analysis is started every time a fastq file is generated. The whole sequencing process lasted 8 h, and the sequencing data were analyzed using a real-time bioinformatics analysis pipeline.

### 2.4. Bioinformatics Analysis Pipeline and Bacterial Detection

Basecalling and quality control of raw data sequencing were performed in a high accuracy mode, using Guppy (v. 3.1.5). Reads with undesired lengths (<200 or >2000 nt) were discarded. The clean reads were then mapped to the 16S rRNA reference gene databases collected from the NCBI database. The E-value of BLAST was set to 1e−5, and the taxonomy of each read was assigned according to the taxonomic information of the mapped subject reference with the highest identity and coverage. A consensus sequence of the reads assigned to the same species was generated by Medaka (v. 0.10.1), and the consensus was remapped to the reference database. The species-level taxonomy of the mapped subject reference was detected as the final result. The bacterial detection of the clinical sample was interpreted according to a strict set of rules described elsewhere [34].

### 2.5. Statistical Analysis

R software (version 3.5.1) was used to analyze the community composition and richness of species. The histogram and heat map based on the relative abundance of species at each classification level were drawn using the *ggplot2* and *pheatmap* packages in R [35,36]. Alpha diversity was described when analyzing the complexity of species diversity using the Chao1 [37] and Shannon indices [38]. If there were only two groups, Student’s t-test was used, but if there were more than two groups, either the Kruskal–Wallis-test or one-way ANOVA was used. Principal Co-ordinates Analysis (PCoA) based on Bray–Curtis distance was performed to obtain principal coordinates and visualize complex and multidimensional data using the *vegan* package in R [39]. A linear discriminant analysis (LDA) effect size (LEfSe) analysis was used to find the biomarker of each group [40]. A non-parametric factorial Kruskal–Wallis-test was used to detect species with significant differences in abundance among different groups, and a Wilcoxon-rank-sum-test was used to judge the difference between the two groups. Finally, LDA analysis was used to evaluate the impact of significant species (LDA score) by setting LDA score ≥2 and obtaining the biomarkers in different groups.

## 3. Results

### 3.1. Patient and Breast Tissue Characteristics

We analyzed a total of 44 breast tissue samples, including benign tissues (*n* = 8) from women with benign tumors of the breast, and breast cancer tissues (*n* = 18), as well as adjacent normal breast tissue pairs (*n* = 18) from women with stage I-IV breast cancer, which were all evaluated and confirmed by pathologists from Renmin Hospital of Wuhan University. The patient demographic and tissue characteristics of the breast tissue samples are shown in Table 1. The average age of breast cancer patients was significantly higher than those in the benign group (*p* < 0.0001, Student’s t-test), and breast cancer tissues were classified by two major breast tumor subtypes based on hormone-receptor expression: ER/PR positive (*n* = 9) and ER/PR negative (*n* = 8). Tumor hormone-receptor status was missing in one breast cancer tissue specimen. Clinical-pathologic characteristics of the patients are listed in Appendix A.

### 3.2. Breast Tissue Microbiome Composition in Breast Tumor/Adjacent Normal/Benign Tissues

A total of 1,443,997 raw operon reads were obtained from all samples (see Appendix A). A total of 1,441,486 clean reads were then generated after processing the raw data (quality control and size selection for 200–2000 nt lengths). Taxonomic classification was carried out on the clean reads that passed processing filters, and the low abundance data of single mapped reads were discarded when considering assigned taxonomic units in order to reduce spurious taxonomic units [41]. Nearly 900 bacterial species were detected overall for all the breast tissue samples (see Appendix A). Proteobacteria, Firmicutes, Actinobacteria, and Bacteroidetes accounted for the most abundant bacterial population at the phylum level (see Appendix A), which was consistent with the results of previous studies [15,42]. At the family level, *Burkholderiaceae* accounted for the major bulk of bacteria followed by *Sphingomonadaceae* and *Alcaligenaceae* across all the breast tissue samples, in that order (Figure 1A). At the genus level, the genus *Ralstonia* was the most dominant bacterial genus prevalent in all tissues (Figure 1B). The genera Sphingomonas and Achromobacter were also abundant in all the breast tissue collected in this study (Figure 1B). We then focused on the distribution of bacteria at the species level. The bacterial composition of each group was similar, with major differences mainly in rare and/or less abundant lineages (Figure 1C). Interestingly, despite being derived from the same donor, the bacterial composition of breast cancer tissue differed from that of adjacent normal tissue (Figure 1C). The bacterium with the highest abundance in all breast tissues was *Ralstonia pickettii*, with the relative abundance increasing with decreasing malignancy (24.58% in tumor vs. 28.60% in adjacent normal vs. 41.51% in benign) (Figure 1C). On the contrary, *Leptothrix mobilis* was more abundant in breast tumors and adjacent tissues (4.17% in tumor vs. 2.01% in adjacent normal vs. 1.77% in benign) (Figure 1C). Other fluctuating microorganisms also merit further attention, and changes in their relative abundance may have a potential impact on breast tissue, which we have included in Appendix A (first 20 species). In addition, we found changes in the relative abundance of some previously reported pathogens under different classifications, such as *Pseudomonas aeruginosa*, which was highly enriched in cancer tissues and was hardly detected in normal and benign tissues (Figure 1C). In Figure 1D, the bar plots illustrate the relative abundance of microbial communities at the species level of all samples in the study, and show the noticeable individual differences in relative bacterial abundance between different groups. A cluster analysis of species abundance in each group was carried out using a heatmap (Figure 1E). *Zhizhongheella caldifontis* and *Verticiella sediminum* were specifically enriched in benign tissues, while enrichment of other bacteria species was found in tumor and adjacent normal tissues, such as *Yonghaparkia alkaliphila* and *Variovorax ginsengisoli* (Figure 1E).

To evaluate microbiome differences between different tissue types, alpha diversity analysis was employed and the results showed that there was no significant difference in Shannon and Chao1 indices between tumor, adjacent normal, and benign tissues (Figure 2A,B). The differences in the composition of microbial communities were visualized using principal coordinate analysis (PCoA) ordination of Bray–Curtis distances. It did not reveal any significant differences in beta diversity, although differences in the microbiome between benign and malignant disease states did exist (Figure 2C). Next, we conducted linear discriminant analysis effect size (LEfSe) analysis to evaluate the microbiome differences between tumor, adjacent normal, and benign tissues, which uncovered different microbiome compositions and identified significant cancer-related biomarkers [14]. The histogram showed that *Sphingobium limneticum*, *Ralstonia solanacearum*, and *Mesorhizobium huakuii* were most abundant in benign samples (Linear Discriminant Analysis (LDA > 2), while *Staphylococcus pasteuri*, *Staphylococcus warneri*, and *Acidovorax temperans* were abundant in normal pairs (LDA > 2) (Figure 3A). Notably, order *Corynebacteriales* including family *Corynebacteriaceae* and genus *Corynebacterium* was enriched in tumor tissues (LDA > 3); the genus *Janthinobacterium* including *Janthinobacterium lividum* was also abundant in tumor tissues (LDA > 3) (Figure 3A). The biomarkers which had relative abundance differences between tumor and other different types of tissue arere presented in Figure 3B-E. Further investigation of these bacteria may be helpful to reveal their influence on breast cancer.

### 3.3. Breast Tissue Microbiome Composition with Different Hormone-Receptor Statuses

Recent studies have revealed that microbiota in the breast may originate from the skin, gut, or mouth through a variety of mechanisms [9,43]. The gut microbiome has long been involved in development and progression of breast cancer through several pathways, including by altering estrogen levels throughout the body and regulating the host’s immune system and tumor immunity [44]. Therefore, we hypothesized that under different hormone-receptor statuses, some microorganisms might be distinguished which have potential effects on breast cancer through a multitude of pathways, similar to gut microbes. In our study, 17 cancer samples were stratified and further examined based on hormone-receptor expression (one patient was excluded due to incomplete data). The results showed that the overall microbiota of breast tissue between the two states appeared similar. Alpha diversity analysis showed no significant differences in either the Shannon index or Chao1 index (Figure 4A,B), and the PCoA analysis used to reflect the beta diversity revealed that the microbial community in hormone- receptor-positive breast cancer was not different from that in hormone-receptor-negative tumors (Figure 4C). Nevertheless, we found multiple bacterial taxa whose prevalence was different between the subtypes. *Ralstonia pickettii* still dominated in both types of breast cancer, but its relative abundance was reduced in hormone-receptor-positive breast tissues (19.03% in hormone-receptor-positive vs. 25.83% in hormone receptor-negative) (Figure 5A). On the contrary, *Curvibacter gracilis* was lower in hormone-receptor-negative breast tissue (7.42% in hormone-receptor-positive vs. 3.54% in hormone-receptor-negative) (Figure 5A). Utilizing a LEfSe analysis to assess differential taxa between the two breast cancer states demonstrated a significantly increased relative abundance in the following species in hormone-receptor-positive breast tissues: Order *Pseudomonadales* (LDA > 4), *Bradyrhizobium betae*, *Acinetobacter johnsonii*, *Bradyrhizobium rifense*, genus *Acinetobacter*, family *Moraxellaceae*, *Undibacterium pigrum*, and genus *Undibacterium* (LDA > 3) (Figure 5B). The histogram shows differences in the quantity of these species between the two disease states (Figure 5C–H). Finally, the heatmap demonstrates elevated *Undibacterium pigrum* in hormone-receptor-positive breast tissues (Figure 5I).

## 4. Discussion

Recently, the influence of microorganisms on cancer susceptibility and progression has attracted great attention in the study of cancer, especially in gastrointestinal tumors, but whether changes in microbiota could cause or affect breast cancer remains unclear. In this study, we used near full-length 16S rRNA operon reads to assess the microbial composition of breast tissues and distinguish the bacterial differences in benign, adjacent normal, and tumor tissues accurately to the species level for the first time. Our results indicated distinct microbial communities in tumor and adjacent normal tissues compared to benign tissues, and compared differences in the microbiota between the tumor tissues with different hormone-receptor statuses. Exploration of the core microbial community in breast tissue and microbial dysbiosis in association with benign and malignant diseases will facilitate an understanding of the bacterial impact on breast health and provide advancements in the diagnosis and treatment for early intervention in breast cancer.

Increasing evidence has established the existence of a unique microbiota in human breast tissue over the past decade. Given the nutrient-rich fatty content of the breast and the widespread location of lobules and ducts leading from the nipple to the mammary gland, Urbaniak et al. suggested that bacteria might be widely distributed in the mammary gland, irrespective of lactation [10,43]. A number of studies have since confirmed this speculation by using next-generation sequencing (NGS) and culture methods [12,13,45,46,47]. Nejman et al. reported that breast tumors had a richer and more diverse microbiome than all other tumor types tested after studying 1526 tumors and their adjacent normal tissues across seven cancer types [11]. Hieken et al. explored the microbiome of aseptically collected breast tissue in benign and malignant disease, and they found that the major differences were mainly in rare and/or less abundant lineages since the difference is not significant using the weighted UniFrac distance [48]. Their investigation demonstrated increased relative abundance in the following low-abundant genera in the breast tissue: *Fusobacterium, Atopobium, Hydrogenophaga, Gluconacetobacter,* and *Lactobacillus*. Genus *Fusobacterium*, especially *Fusobacterium nucleatum*, has already been confirmed to act by secreting virulence factors as well as creating a pro-inflammatory environment that promotes carcinogenesis in colon cancer [49,50], and a recent study has also found its possible impact on breast cancer progression [20]. Meng et al. compared microbiome profiles in the breast tissues collected from Chinese patients with benign and malignant disease and identified characteristic microbial biomarkers [42]. Although alpha diversity analysis revealed that there was no significant difference in Shannon index, and the generated weighted and unweighted UniFrac distance metric also did not show significant differences in beta diversity, the microbial composition of benign and malignant disease states did differ at the phylum and family levels [42], consistent with our findings. They found the enriched microbial biomarkers in tumor tissues included genus *Propionicimonas* and families *Micrococcaceae*, *Caulobacteraceae*, etc. [42] Unfortunately, these investigations using NGS are still in their infancy and lack characteristic microbial biomarkers identified at the species level.

One major finding from our study was the accurate identification of a distinct microbiota associated with benign and malignant disease based on higher species resolution. At the genus level, the bacterium *Ralstonia* was the most dominant bacterial genus prevalent in the breast. Smith et al. stated that breast tissues from non-Hispanic Black (NHB) women had a higher abundance of genus *Ralstonia* compared to non-Hispanic White (NHW) tumors [14], but a recent study reported that *Ralstonia* was the dominant genus with no significant differences in abundance between tumor vs. normal tissues [15]. Possible reasons for the inconsistency of these studies include different eating habits, living environments, and metabolic levels in different races, but there is no doubt that *Ralstonia* does exist abundantly in the breast tissue, and its possible role in the incidence of breast cancer needs to be further explored by researchers. In our study, we identified that the dominant species is *Ralstonia pickettii*, of which the relative abundance was higher in benign tissues compared to tumor tissues. It is a non-fermentative, Gram-negative bacterium from the *Ralstonia* genus, which is emerging as one kind of opportunistic pathogen closely associated with nosocomial infections [51,52]. Multiple nosocomial outbreaks have been caused by medical solutions contaminated with *R. pickettii* [53,54]. However, the involvement of this bacterium in the carcinogenesis of any organ has not been thoroughly studied. In the study by Higuchi et al. [55], *Streptococcus australis* and *R. pickettii* were identified in almost all mesothelioma patients, but it is unclear whether the bacterial profiles with high levels of bacterial composition and abundance are causally associated with oncogenesis. Future studies with larger sample sizes are needed to determine their potential association with tumorigenesis and to explore the underlying mechanisms.

We further examined the breast tissue microbiome composition with different hormone-receptor statuses. Considering that hormone-receptor-positive breast cancer is closely related to hormone secretion, and it has long been proposed that the gut microbiome contributes to breast carcinogenesis by modifying systemic estrogen levels [56,57], we therefore saw the necessity of identifying unique microbial signatures associated with different hormone-receptor statuses, which may play a possibly vital role in local estrogen metabolism and immune regulation similar to that of the gut microbiome. Considering that the research on the relationship between progesterone and the microbiome is still in its infancy, and ER/PR expression was consistent in all samples collected in this study, we hereby refrain from further analysis of the role of progesterone receptors alone. Interestingly, the specific bacterial markers we identified were all concentrated in hormone-receptor-positive breast cancer. Wang H et al. has reported a distinct microbiome in breast cancer with different hormone-receptor statuses, and suggested that *Methylobacterium* was significantly enriched in hormone-receptor-positive breast cancer [12]. In our study, we observed that order *Pseudomonadales* was abundant in hormone-receptor-positive breast tissue. Order *Pseudomonadales,* especially genus *Pseudomonas*, is widespread in the breast with low proportional abundance [10,15]. In the mouth and vagina, it has been found to be significantly enriched in women without intraepithelial lesions compared to women with squamous intra-epithelial neoplasia [58], but little is known regarding the role of *Pseudomonas* in breast carcinogenesis. In our study, we found changes in the relative abundance of *P. aeruginosa*; it was highly enriched in cancer tissues, especially in hormone-receptor-positive tumor tissues, and was hardly detected in normal and benign tissues. It is widely known that pathogen infection is one of the severe complications of malignant tumors, and it has been reported that cancer patients undergoing chemotherapy are at high risk of becoming infected with multi-drug-resistant *P. aeruginosa* [59], which was also identified in our analysis. However, Chiba et al. [60] reported that neoadjuvant chemotherapy increased the proportional abundance of *Pseudomonas* to 85%, suggesting chemotherapy induced its preferential growth or survival. They also mentioned that treatment of breast cancer cells with *P. aeruginosa*-conditioned media differentially effected proliferation and modulated doxorubicin-mediated cell death, suggesting that *P. aeruginosa* may be related to breast carcinogenesis, but the specific mechanisms underlying these effects are yet to be clarified [60]. In addition, *Undibacterium pigrum* was also found to be abundant in hormone-receptor-positive breast tissue. This is a Gram-negative bacterium of the genus *Undibacterium,* which can be isolated from purified water, but unfortunately, its pathogenicity in humans is not yet known [61]. Additional studies will aid in enlightening our observations and further exploring the potential impact of the breast microbiome on cancer development and progression.

ONT nanopore sequencing technology was commercially released in 2014, and since then has become suitable for multiple applications, such as pathogen identification [62], genome assembly [63], and antibiotic-resistance gene detection [64]. Significant advances have been achieved in both the read quantity and accuracy of nanopore sequencing on account of improvements in sequencing chemistry/pore design and better algorithms for basecalling [65]. With the introduction of the Guppy algorithm, ONT indicates a median read accuracy of 89–94% from microbial genomes [66,67]. In addition, the improvements in sequence accuracy from 60% to >90% are illustrated by reason of the changes from R6 to R9 chemistry [68], and the single-read accuracy has increased to >95% for the latest R10.3 flow cells [69,70]. Due to the advantage of generating long reads to allow sequencing of the entire 16S rRNA gene, nanopore sequencing was believed to provide high-resolution results regarding bacterial classification. As one of the earliest reports, Shin et al. used nanopore sequencing to sequence full-length 16S rRNA amplicons from the mouse gut microbiota [41]. They found that both sequencing data were highly similar at all taxonomic resolutions, but at the species level, nanopore sequencing allowed the identification of more species than short-read sequencing. Wei et al. generated 16S rRNA sequences with the same batch of total genomic DNA extracted from fecal samples using Illumina and nanopore technology platforms [31]. Their results suggested that sequencing of the entire 16S rRNA gene improves the accuracy of microbial community classification at both genus and specific levels, compared with the sequencing of the 16S rRNA gene intra-variable region [31]. Taylor et al. compared nanopore sequencing with meta-transcriptomics and Illumina sequencing for microbiome analysis of colorectal tumor tissues [32]. The results showed that long-read sequences could compensate for low read depth for classification purposes [32]. In a word, applying long-read sequencing approaches to study 16S rRNA genes will provide new possibilities for the identification of precise bacterial species in a great variety of microbial compositions.

One drawback of this study was the small sample size. Larger investigations are needed to reduce the within-group differences and further confirm our conclusion. In addition, studies with larger sample sizes and in-depth mechanism analysis are also warranted to systematically elaborate the relationship between breast tissue microbiomes and other indicators strongly associated with breast cancer, such as epidermal growth factor receptor 2 (HER2), and their impact on breast cancer. Another weakness of the present study is the limited demographic data. As a consequence, our analysis may be potentially confounded by the effects of influencing factors such as age and menopausal status, and we presently do not know the exact effect of these confounding factors on the breast microbiota. Furthermore, since the microbiota composition was determined based on the nanopore sequencing data obtained with the taxonomy-supervised analysis, which allocates sequences directly into taxonomic bins instead of using computational operational taxonomic units (OTU) clustering based on sequence similarity, several OTU based analyses cannot be achieved in this study, such as phylogenetic relationship construction and functional predictive analysis, etc. Although improvements for nanopore sequencing have been made in terms of accuracy and data analysis, other sequencing methods including shotgun metagenomics and transcriptomic RNA-seq technology should be employed in further research to reveal deeper characterization, such as the functional genes in the cancer microbiome, and confirm the microbial species discovered in this study. Further investigations with a larger patient cohort on the microflora composition in the mammary gland may be of great help in understanding the multi-faceted roles of the microbiome in breast cancer.

## 5. Conclusions

In summary, we showed for the first time the differences in the microbiome of human breast tissue with different degrees of malignancy at the species level using Nanopore sequencing. Specially, we compared the microbial composition in breast cancer tissues with different hormone-receptor statuses. It remains to be elucidated whether the identified bacterial profiles are causally associated with carcinogenesis, or merely reflective of the pathological processes in breast cancer. Furthermore, it is currently unclear whether it is a single organism driving the promotion of carcinogenesis, or an interplay of polymicrobial interactions. Further investigation of the potential role of the tissue microbiome in the development of breast cancer is warranted to characterize a microbial risk signature and develop potential microbial-based prevention therapies.

## Figures and Tables

**Figure 1 jpm-13-00174-f001:**
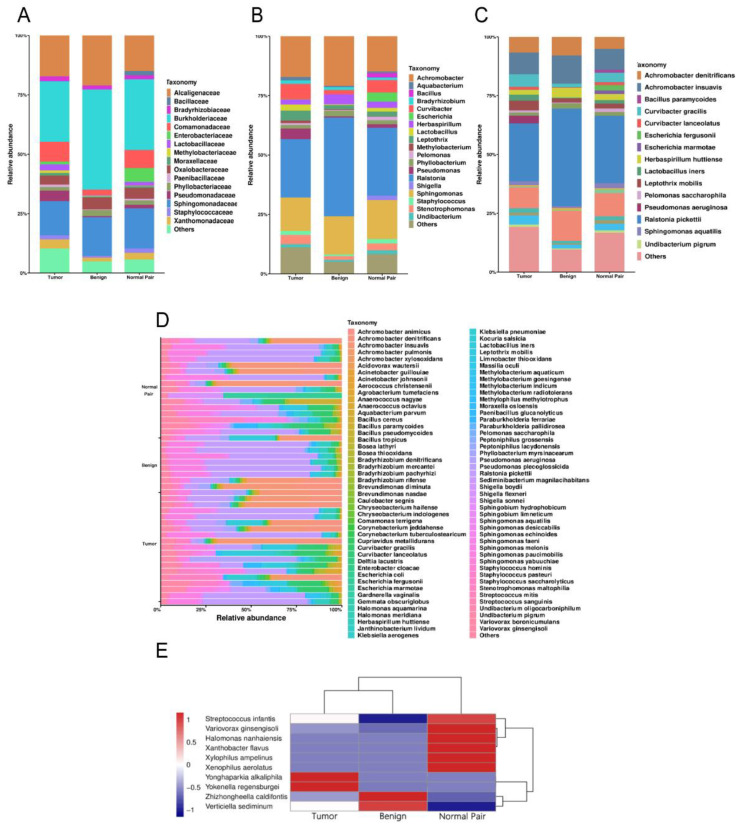
(**A**) Bar plots of the taxonomic profiles of the benign (*n* = 8), normal pair (*n* = 18), and breast tumor tissue (*n* = 18) microbiota at the family level, (**B**) genus level, and (**C**) species level with a relative abundance >1% are shown. (**D**) Relative abundance of bacterial species within breast tissue samples across all tissues. Each bar represents a subject and each colored box a bacterial taxon, the height of each colored box represents the relative abundance of that organism within the sample. (**E**) A heatmap illustrates the cluster analysis of species abundance in each group. A blue color represents rare or absent species while a red color represents abundant species.

**Figure 2 jpm-13-00174-f002:**
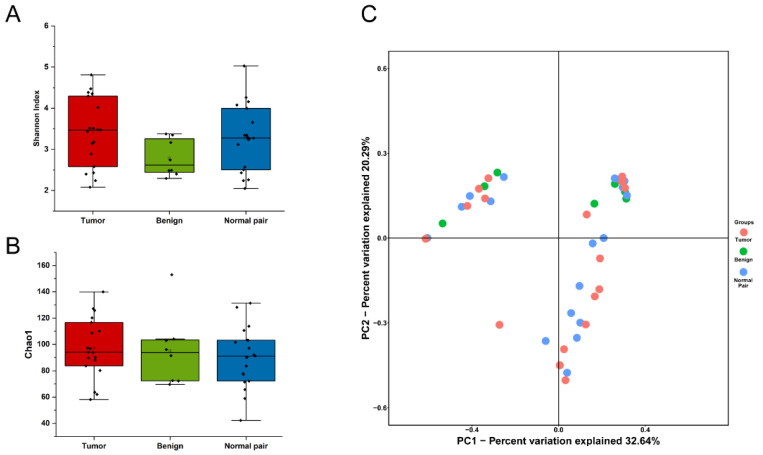
(**A**) Boxplot comparing Shannon index and (**B**) Chao1 between benign, normal pair, and breast tumor tissues (Shannon index, *p* = 0.171; Chao1, *p* = 0.645, one-way ANOVA). (**C**) PCoA plots show the clustering pattern of the three groups based on Bray–Curtis distance (*p* = 0.699, Adonis).

**Figure 3 jpm-13-00174-f003:**
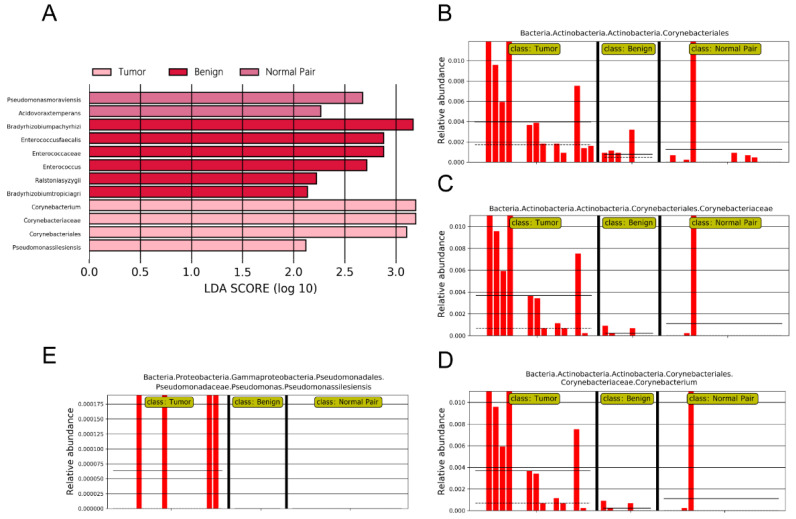
(**A**) LDA scores predict microbiota associated with benign, normal pair, and breast tumor tissue microbiomes. (**B**–**E**) Bar plots show the potential biomarkers which had relative abundance differences between tumor and other different types of tissues. Each bar represents a sample.

**Figure 4 jpm-13-00174-f004:**
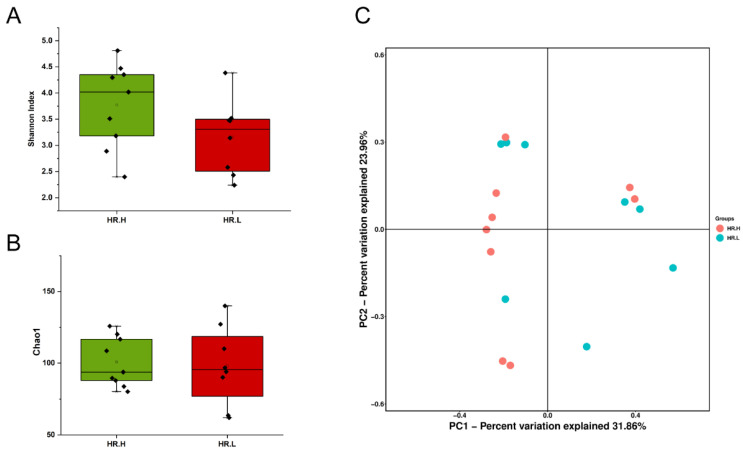
(**A**) Boxplot comparing Shannon index and (**B**) Chao1 between hormone-receptor-positive tumors and hormone-receptor-negative tumors (Shannon index, *p* = 0.121; Chao1, *p* = 0.805, Student’s t test). (**C**) PCoA plots show the clustering pattern of the two groups based on Bray–Curtis distance (*p* = 0.687, Adonis). HR.H, hormone-receptor-positive tumors; HR.L, hormone-receptor-negative tumors.

**Figure 5 jpm-13-00174-f005:**
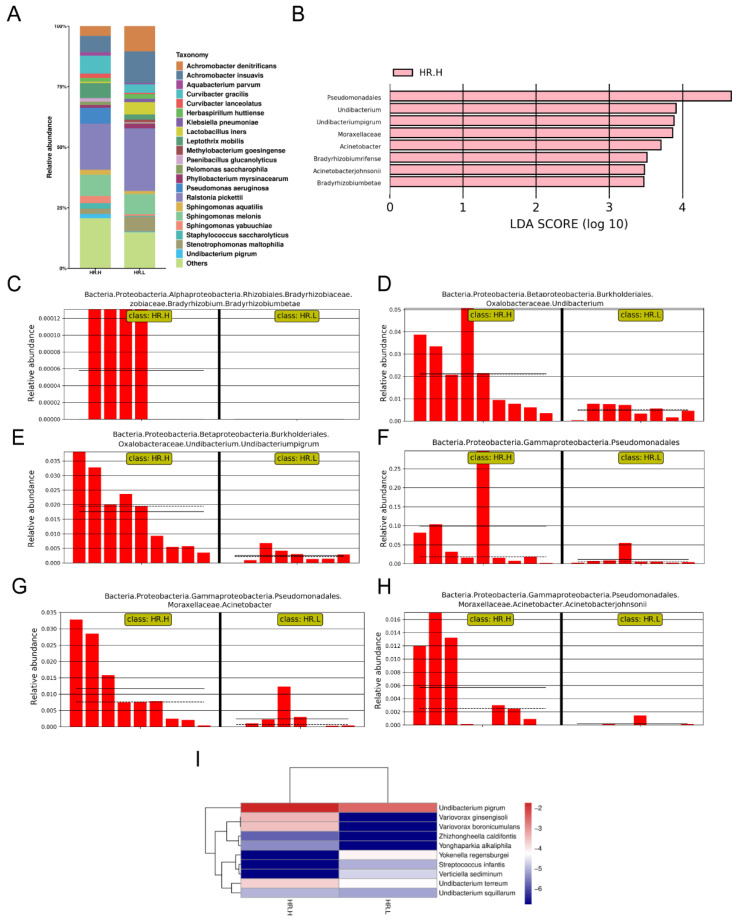
(**A**) Bar plots of the taxonomic profiles of hormone-receptor-positive tumors (*n* = 9) and hormone-receptor-negative tumors (*n* = 8). Microbiota at species level with a relative abundance >1% are shown. (**B**) LDA scores computed for the two groups. (**C**–**H**) Bar plots show the abundance of the potential biomarkers between the two groups. Each bar represents a sample. (**I**) A heatmap illustrating the cluster analysis of species abundance in each group. Blue color represents rare or absent species while red color represents abundant species. HR.H, hormone-receptor-positive tumors; HR.L, hormone-receptor-negative tumors.

**Table 1 jpm-13-00174-t001:** Demographics of study patients.

Variable	Tumor	Benign	Total
Mean Age, years Average (range) Menopausal Stapotus Premenopausal Postmenopausal Missing Stage 1 2 3/4 Missing ER/PR Positive (+) Negative (−) Missing HER2 Positive (+) Negative (−) Missing	54 (43–90) 3 15 0 3 8 5 1 9 8 1 13 4 1	25 (18–35) 8 0 0 NA NA NA NA NA NA NA NA NA NA	45 (18–90) 11 15 0 NA NA NA NA NA NA NA NA NA NA

## Data Availability

All sequencing data associated with this study can be accessed at the NCBI Sequence Read Archives under BioProject accession No. PRJNA769561, https://www.ncbi.nlm.nih.gov/bioproject/PRJNA769561/ (accessed on 13 October 2021).

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
