# Peer review of "Species-Level Characterization of the Microbiome in Breast Tissues with Different Malignancy and Hormone-Receptor Statuses Using Nanopore Sequencing"

_jpm, 2023, doi:10.3390/jpm13020174_

Round 1

Reviewer 1 Report

Manuscript Title: Species-level Characterization of the Microbiome in Breast Tissues with Different Malignancy and Hormone Receptor Statuses Using Nanopore Sequencing

Manuscript Summary: In the manuscript, Luo,L et al. investigated the differential composition in microbiome species between benign and malignant breast tissue. The authors report in the manuscript that 900 different bacterial species were identified in the breast tissue using Oxford Nanopore long-read sequencing within the 4 dominant phyla: Proteobacteria, Firmicutes, Actinobacteria, and Bacteriodetes. The highest abundant bacteria in breast tissue was identified to be Ralstonia pickettii which decreased with increasing malignancy. In addition, the authors also report the association between increases in abundance of Pseudomonas in hormone receptor-positive cancer. The study findings presented in the study are interesting as there is growing interest in the field regarding the role of microbiome in regulating tumor progression. To have an association in breast microbiome and breast cancer development and progression will also be a great addition to understanding more on the pathophysiology of breast cancer. Therefore, the findings presented in the study is off substantial scientific merit. I would like to recommend a few additional comments to the authors to consider as revisions for this manuscript so as to increase its scientific rigor.

Minor Concerns

1.       Figure 1D, taxonomy titles are so small and are difficult to read. Please consider increasing the font size of the legends.

2.       The authors talk on the discovery of poly-microbiome in the breast tissue and its potential role as risk factor for breast cancer development. However, the authors have not discussed or postulated on how microbiome can affect breast cancer development. It would be great if the authors can include a couple of sentences in the Introduction section talking on the implication of microbiome on cancer development to justify the significance of the findings presented in the manuscript.

3.       When the authors talk about “adjacent normal tissue,” it is important to specify what the terminology is referring to. Are the authors in this scenario referring to the breast cancer stroma or non-malignant breast epithelium?

4.       As a follow up to point 3 above, how did the authors make this distinction in spatial enrichment in breast tissue if all tissue samples were cut into small pieces and homogenized followed up with DNA extraction (line 99 to 101 in Materials and Methods section).

5.       The authors have associated differential composition in microbiome with respect of Estrogen receptor status. It is important for the authors to include in the discussion section if progesterone and HER2 receptor status was also considered for similar analysis or alternatively why was it excluded.

Author Response

Thank you for your positive evaluation of our work. Those precious comments and advice are all valuable and very helpful for revising and improving our paper. We have revised the manuscript accordingly.

Q1: Figure 1D, taxonomy titles are so small and are difficult to read. Please consider increasing the font size of the legends.

Response: We have enlarged the font of the legends in the figure 1D in order to make it easier to read.

Q2: The authors talk on the discovery of poly-microbiome in the breast tissue and its potential role as risk factor for breast cancer development. However, the authors have not discussed or postulated on how microbiome can affect breast cancer development. It would be great if the authors can include a couple of sentences in the Introduction section talking on the implication of microbiome on cancer development to justify the significance of the findings presented in the manuscript.

Response: We deeply appreciate the reviewer’s suggestion. According to the reviewer’s comment, we have added a more detailed description in regard to the implication of microbiome on cancer development into the Introduction section (Line 44-54, Page 2).

Q3: When the authors talk about “adjacent normal tissue,” it is important to specify what the terminology is referring to. Are the authors in this scenario referring to the breast cancer stroma or non-malignant breast epithelium?

Response: Thank you for underlining this deficiency. As a matter of fact, “adjacent normal tissue” is defined as non-malignant breast tissue that was up to 5 cm away from the edge of the tumor and was evaluated and confirmed by a pathologist to be histologically free of any tumor cells or lesions. We have added the information into the Method section to clarify this point (Line 112-116, Page 3).

Q4: As a follow-up to point 3 above, how did the authors make this distinction in spatial enrichment in breast tissue if all tissue samples were cut into small pieces and homogenized followed up with DNA extraction (line 99 to 101 in Materials and Methods section).

Response: Thank you for pointing this out. All tissue samples were grouped and numbered immediately when aseptically collected in the operating room, and the DNA was extracted in batches according to the unique group and number to distinguish each tissue sample during the extraction process. We apologize for being misleading in these parts of our paper, and have already emphasize this point in the article. (line 120-121, page 3)

Q5: The authors have associated differential composition in microbiome with respect of Estrogen receptor status. It is important for the authors to include in the discussion section if progesterone and HER2 receptor status was also considered for similar analysis or alternatively why was it excluded.

Response: Thank you for your careful review. The progesterone receptor and epidermal growth factor receptor 2 (HER2) are both important prognostic markers and therapeutic targets of breast cancer, but due to the small sample size and insufficient literature, we have not been able to adequately discuss these two aspects. Detailed reasons are as follows:

In regard to the expression of HER2, the number of patients with the negative expression of HER2 is far less than the count of patients with the positive expression on account of the small sample size (4:13) (see table 1), which might result in bias in the subgroup analyses. As a consequence, the relationship between HER2 receptor expression and intratumoral microbiomes has not been further discussed in our study.

As for the progesterone receptor, we did not pursue the discussion for two main reasons:

1. As is known to all, the PR is encoded by an estrogen-regulated gene, and its synthesis requires estrogen and ER; therefore, ER-positive tumors are commonly PR positive, whereas ER-negative tumors are usually PR negative (Ref1). In our study, ER/PR expression was consistent in all samples collected, so we conducted a joint analysis, namely, the relationship between “hormone receptor status” and microbiome. After analyzing the full text, we found that the legends in FIG. 4 and FIG. 5 only indicated estrogen (ER.L/ER.H), but did not indicate hormone-related receptors. Hence we modified the legends in the two figures in the hope that such modification could reduce readers' misunderstanding (HR.L/HR.H).

2. At present, the research on the relationship between progesterone and microbiome is still in its infancy; however, Estrogen is better known for its association with intestinal bacteria and breast cancer. The intestinal bacteria have long been involved in development and progression of breast cancer through several pathways including altering estrogen levels throughout the body and regulation of the host’s immune system and tumor immunity (Ref2). Previous studies have suggested that parts of the breast microbiome may come from the gut (Ref3), so we hypothesized that the breast microbiome may play multiple roles in promoting breast cancer development, such as local estrogen metabolism, thereby promoting the development of hormone-receptor-positive breast cancer. On the contrary, the effect of progesterone is usually based on the effect of estrogen, and the role of progesterone in breast cancer is still controversial. Therefore, we focused on the relationship between estrogen and bacteria and breast cancer in the discussion section.

We sincerely hope that future studies with larger sample sizes and in-depth mechanism analysis can systematically elaborate the relationship between estrogen, progesterone, HER receptor and breast microbiomes, and their impact on breast cancer. We sincerely thank the reviewer for the suggestion. To further clarify this, we have added several sentences in the Discussion section, see line 381-384, page 15; line 439-443, page 16).

Ref:

  1. Li Y, Yang D, Yin X, et al. Clinicopathological Characteristics and Breast Cancer-Specific Survival of Patients With Single Hormone Receptor-Positive Breast Cancer. JAMA Netw Open. 2020 Jan 3;3(1):e1918160. doi: 10.1001/jamanetworkopen.2019.18160.
  2. Ervin SM, Li H, Lim L, et al. Gut microbial β-glucuronidases reactivate estrogens as components of the estrobolome that reactivate estrogens. J Biol Chem. 2019 Dec 6;294(49):18586-18599. doi: 10.1074/jbc.RA119.010950.
  3. Urbaniak C, Burton JP, Reid G. Breast, milk and microbes: a complex relationship that does not end with lactation. Womens Health (Lond). 2012 Jul;8(4):385-98. doi: 10.2217/whe.12.23

Reviewer 2 Report

Luo et al. characterized and identified the microbiome compositions in breast tissues with different malignancy at the species level. By sequencing the microbial 16S rRNA using Oxford Nanopore Sequencing, they successfully identified around 900 bacterial species. By computational analysis of the long reads, they identified several bacterial species that have differential abundance in the three groups of breast tissues. For example, they found that Ralstonia pickettii is the most abundant bacterial species and has higher abundance in benign breast tissue compared to malignant tumor and adjacent normal tissue. In addition, they also compared the bacterial composition between hormone receptor positive and hormone receptor negative tumors and provided candidate biomarkers for different groups of breast tissues. Their work adds to our knowledge of the microbiome composition in breast tissues at the species level and will help in-depth studies of the functions of individual bacterial species in breast carcinogenesis.

Minor comments:

1.      Line 119, “16S rDNA reference gene” is used, while 16S rRNA gene is used in the text above. The name of the gene should be consistent.

2.      Figure S1 and S2, the figure legend needs more information to explain what the samples are. It is mentioned in Table S1 that samples L1-L8 are benign tumors and samples C1-19/P1-P19 are malignant tumor and adjacent normal tissue. This information should be included in the legend of Figure S1 and S2 to make things clearer.

3.      Line 173 and 174, the sentence “[t]he enrichment of bacterial genera Sphingomonas and Achromobacter” is confusing. The word “enrichment” is not appropriate since the bacteria composition is not compared with the bacteria composition in other tissues etc.

4.      The order of the three groups of breast tissue samples in the figure panels keeps changing throughout the paper. It is beneficial to keep the order of the samples the same in all figures, for example, in the order of decreasing or increasing malignancy to better explain the results.

5.      Line 220, the sentence should refer to Figure 3A, not Figure 2A.

6.      Figure 3B-E, the font in the figure is too small.

7.      Line 257, the sentence should refer to Figure 5A, not Figure 4A.

8.      Figure 5A-H, the font in the figure is too small.

Author Response

Thanks very much for taking your time to review this manuscript. We really appreciate all your generous comments and suggestions that has helped to make our study clearer and more comprehensive.

Q1: Line 119, “16S rDNA reference gene” is used, while 16S rRNA gene is used in the text above. The name of the gene should be consistent.

Response: Thank you for pointing this out. We feel sorry for our carelessness and have corrected this error to keep the name of the gene consistent throughout the article.

Q2: Figure S1 and S2, the figure legend needs more information to explain what the samples are. It is mentioned in Table S1 that samples L1-L8 are benign tumors and samples C1-19/P1-P19 are malignant tumor and adjacent normal tissue. This information should be included in the legend of Figure S1 and S2 to make things clearer.

Response: We are grateful for the suggestion. To be clearer and in accordance with the reviewer concerns, we have added more detailed interpretations to the legend of Figure S1 and S2 as follows (see line 472-478, page 16-17).

Q3: Line 173 and 174, the sentence “[t]he enrichment of bacterial genera Sphingomonas and Achromobacter” is confusing. The word “enrichment” is not appropriate since the bacteria composition is not compared with the bacteria composition in other tissues etc.

Response: We regret for the inappropriate expression in the article and re-wrote the sentence in the revised manuscript as the following (line 193-196, page 5):

“The genera Sphingomonas and Achromobacter were also abundant in all the breast tissues collected in this study.”

Q4: The order of the three groups of breast tissue samples in the figure panels keeps changing throughout the paper. It is beneficial to keep the order of the samples the same in all figures, for example, in the order of decreasing or increasing malignancy to better explain the results.

Response: We agree with the comment and have adjusted the order of the sample groups in all the figures to make it easier for readers to understand. In Figure1-3, the groups are listed in the following order: Tumor, Benign, Normal Pair; In Figure4-5, the order of the groups is: HR.H; HR.L.

Q5: Line 220, the sentence should refer to Figure 3A, not Figure 2A.

Q6: Figure 3B-E, the font in the figure is too small.

Q7: Line 257, the sentence should refer to Figure 5A, not Figure 4A.

Q8: Figure 5A-H, the font in the figure is too small.

Response to Q5 and Q7: Thank you very much for your careful checks. We are really sorry for our careless mistakes and have already made the corrections.

Response to Q6 and Q8: As suggested by the reviewer, we have enlarged the font in the figures (Figure 3B-E and Figure 5A-H).